# Echocardiographic and Electrocardiographic Determinants of Atrial Cardiomyopathy Identify Patients with Atrial Fibrillation at Risk for Left Atrial Thrombogenesis

**DOI:** 10.3390/jcm11051332

**Published:** 2022-02-28

**Authors:** Taiyuan Huang, Schurr Patrick, Louisa Katharina Mayer, Björn Müller-Edenborn, Martin Eichenlaub, Martin Allgeier, Jürgen Allgeier, Heiko Lehrmann, Christoph Ahlgrim, Marius Bohnen, Simon Schoechlin, Dietmar Trenk, Nikolaus Jander, Franz Josef Neumann, Thomas Arentz, Amir Jadidi

**Affiliations:** 1University of Freiburg, Heart Center, Campus Bad Krozingen, Clinics for Cardiology and Angiology II, Division of Cardiac Arrhythmias, 79189 Bad Krozingen, Germany; louisa.katharina.mayer@uniklinik-freiburg.de (L.K.M.); bjoern.mueller-edenborn@uniklinik-freiburg.de (B.M.-E.); martin.eichenlaub@uniklinik-freiburg.de (M.E.); juergen.allgeier@uniklinik-freiburg.de (J.A.); heiko.lehrmann@uniklinik-freiburg.de (H.L.); christoph.ahlgrim@uniklinik-freiburg.de (C.A.); marius.bohnen@uniklinik-freiburg.de (M.B.); simon.schoechlin@uniklinik-freiburg.de (S.S.); franz-josef.neumann@uniklinik-freiburg.de (F.J.N.); thomas.arentz@uniklinik-freiburg.de (T.A.); 2University of Freiburg, Heart Center, Campus Bad Krozingen, Clinics for Cardiology and Angiology II, Division of Cardiac Imaging, 79189 Bad Krozingen, Germany; martin.allgeier@uniklinik-freiburg.de (M.A.); nikolaus.jander@uniklinik-freiburg.de (N.J.); 3University of Freiburg, Heart Center, Campus Bad Krozingen, Clinical Pharmacology, 79189 Bad Krozingen, Germany; dietmar.trenk@uniklinik-freiburg.de

**Keywords:** atrial fibrillation, atrial cardiomyopathy, global longitudinal strain, p-wave analysis, left atrial thrombogenesis, stroke

## Abstract

Objective: Atrial cardiomyopathy (ACM) is associated with development of AF, left atrial (LA) thrombogenesis, and stroke. Diagnosis of ACM is feasible using both echocardiographic LA strain imaging and measurement of the amplified p-wave duration (APWD) in digital 12-lead-ECG. We sought to determine the thresholds of LA global longitudinal strain (LA-GLS) and APWD that identify patients with AF at risk for LA appendage (LAA) thrombogenesis. Methods: One hundred and twenty-eight patients with a history of AF were included. Left atrial appendage maximal flow velocity (LAA-Vel, in TEE), LA-GLS (TTE), and APWD (digital 12-lead-ECG) were measured in all patients. ROC analysis was performed for each method to determine the thresholds for LA-GLS and the APWD, enabling diagnosis of patients with LAA-thrombus. Results: Significant differences in LA-GLS were found during both rhythms (SR and AF) between the thrombus group and control group: LA-GLS in SR: 14.3 ± 7.4% vs. 24.6 ± 9.0%, *p* < 0.001 and in AF: 11.4 ± 4.2% vs. 16.1 ± 5.0%, *p* = 0.045. ROC analysis revealed a threshold of 17.45% for the entire cohort (AUC 0.82, sensitivity: 84.6%, specificity: 63.6%, Negative Predictive Value (NPV): 94.3%) with additional rhythm-specific thresholds: 19.1% in SR and 13.9% in AF, and a threshold of 165 ms for APWD (AUC 0.90, sensitivity: 88.5%, specificity: 75.5%, NPV: 96.2%) as optimal discriminators of LAA-thrombus. Moreover, both LA-GLS and APWD correlated well with the established contractile LA-parameter LAA-Vel in TEE (r = 0.39, *p* < 0.001 and r = −0.39, *p* < 0.001, respectively). Conclusion: LA-GLS and APWD are valuable diagnostic predictors of left atrial thrombogenesis in patients with AF.

## 1. Introduction

Atrial cardiomyopathy (ACM) of the left atrium (LA) is associated with increased fibro-fatty replacement of atrial cardiomyocytes [1,2], electrical heterogeneity, slow conduction, and development of atrial fibrillation (AF) [3,4,5]. In addition, the loss of atrial cardiomyocytes relates to an impairment of left atrial mechanical function with reduced atrial compliance and impaired contractile function [6], both of which are associated with reduced left atrial appendage (LAA) maximal emptying flow velocity (LAA-Vel) in TEE, an increased risk for LA-thrombogenesis, and embolic stroke [7].

We recently demonstrated that both the duration of the amplified digital sinus p-wave in 12-lead ECG (APWD) [8,9], and echocardiographic LA global longitudinal reservoir strain (LA-GLS) in sinus rhythm (SR), enable diagnosis of ACM [10,11], as identified by the presence of left atrial (LA) low-voltage substrate >2 cm^2^ in high-definition voltage mapping during sinus rhythm (SR). Multiple studies revealed that presence of LVS is associated with increased AF recurrence rates after pulmonary vein isolation for AF [3,12,13]. In addition, the clinical relevance of LA low-voltage substrate (LVS; in electro-anatomical voltage mapping) was demonstrated in a multicenter study, revealing a strong association of LVS with a history of stroke in patients with AF [14]. Moreover, recent reports suggest that LA-GLS measured during AF can be used as a predictor for stroke risk in addition to CHA2DS2-VASc-score [14].

Albeit the aforementioned promising diagnostic performances of strain-imaging and APWD with regard to ACM, most of them were reported in SR and their efficacy in prediction of LA-thrombus has not been evaluated yet, and the potential influence of rhythm on LA-GLS is unknown.

In the current study, we evaluate both previously established markers of ACM (APWD and LA-GLS) with regard to their ability to identify AF patients at risk for LA thrombogenesis. Specifically, we aimed to assess the diagnostic thresholds for APWD and LA-GLS that allow identification of patients at risk for LA thrombus formation.

## 2. Methods

### 2.1. Patient Population

One hundred and twenty-eight patients referred to our center for symptomatic AF between 2018 and 2021 were included in the current study. Patients presented for scheduled catheter ablation for AF or for electrical cardioversion in the context of symptomatic AF. All patients underwent TEE prior to electrical cardioversion or for exclusion of LA-thrombus and guidance of transseptal puncture at the beginning of the catheter ablation procedure for AF. General inclusion criteria were: (1) Confirmed diagnosis of AF in previous or current ECG recordings in all patients. (2) Presence of SR in at least one digital 12-lead-ECG recording within 6 months prior to TEE. Exclusion criteria were previous atrial catheter ablation, prior cardiac surgery, decompensated heart failure, or high-degree valvulopathy. From 2018 to 2021 a large cohort of patients (1990 patients) underwent TEE prior to electrical cardioversion. Diagnosis of LAA thrombus was established in 39 of 1990 (1.95%) patients. In 26 of 39 patients with LAA thrombus, digital 12-lead-ECG during SR and LA-GLS were documented. One hundred and two consecutive AF patients without LAA thrombus in TEE served as the control group and were included for comparison of baseline clinical characteristics, as well as echocardiographic and electrocardiographic markers of ACM. The study was approved by the institutional ethics committee and all patients provided written informed consent prior to enrolment.

### 2.2. Transthoracic Echocardiography (TTE)

Standardized TTE was performed in all patients in accordance with current guidelines as described previously [11,15]. LA diameter was measured in parasternal long-axis at end-systole. Left ventricular end-diastolic dimension (LVEDD) was measured in M-mode echocardiograms derived from 2D images in the parasternal long axis. Simpson’s method was used to determine left ventricular ejection fraction (LV-EF). Two-dimensional speckle tracking was analyzed in accordance with the current guidelines [16]. For calculation of the LA global longitudinal strain, apical four- and two-chamber views were used (frame rate between 57 and 90 frames per second). LA global longitudinal strain (LA-GLS) was automatically analyzed in four- and two-chamber views using TomTec software (AutoStrain, TomTec Imaging Systems, Unterschleissheim, Germany): A complete R–R cycle (end-diastole to end-diastole) was automatically selected, and endocardial borders were automatically placed by the software. LA-GLS was measured automatically in the reservoir phase (between mitral valve closure and mitral valve opening) and calculated as an average of three consecutive beats from both two- and four-chamber views (Figure 1).

### 2.3. Transesophageal Echocardiography (TEE)

All TEEs were evaluated for presence of LAA thrombus by two independent cardiologists. The left atrium and the left atrial appendage were inspected closely for the presence of thrombus. The left atrial appendage was visualized in at least two orthogonal planes carefully optimized to avoid shadowing artifacts. Flow velocity signals were obtained 1 cm below the entrance of the left atrial appendage and spectral pulsed-wave Doppler was monitored for 10 s; the maximal left atrial appendage emptying flow velocity over the monitored interval was measured. Thrombus formation was diagnosed in the presence of a solid wall-adherent structure detectable in two orthogonal planes carefully differentiating them from musculi pectinati and other anatomical structures.

### 2.4. Amplified Sinus-P-Wave Duration (APWD)

APWD of each patient was measured using the most recent digital 12-lead-ECG recording in SR (at admission or within the past 6 months) in accordance with previous protocol of amplified measurement [9]. In brief, APWD was measured using digital calipers with a sweep speed of 100–200 mm/s and a scale of 40–80 mm/mV.

## 3. Statistical Analysis

Continuous variables were expressed as mean (standard deviation) or median (interquartile range) depending on distribution. The independent samples *t*-test or Mann–Whitney U test was used to compare continuous variables between two groups; for comparison over two groups, ANOVA testing or an equivalent non-parametric test was performed. Comparison of categorical variables was conducted using the chi-square test or Fisher’s exact test. The receiver-operating characteristic (ROC) curve was performed to determine the optimal thresholds of LA-GLS and APWD to identify patients with LAA-thrombus. Pearson’s correlation analysis was used to assess the correlation between APWD and LA-GLS, LA-GLS with LAA-Vel, and APWD with LAA-Vel. Two-tailed *p* values were calculated for all tests and considered significant at a *p* < 0.05. All statistical analysis was performed with SPSS version 27.0 for Macintosh (IBM Corporation, Armonk, NY, USA) or GraphPadPrism version 9.0 for Macintosh (GraphPad Software, LaJolla, CA, USA).

## 4. Results

### 4.1. Baseline Characteristics

A total of 128 patients were included in the current study (mean age: 65.8 years, 67% male). Eighty-two patients (64.1%) were in SR and 46 patients (35.9%) were in AF during LA-strain imaging. Three of 128 (2.3%) patients had no previous oral anticoagulant (OAC) therapy. Vit-K-antagonists were used in 29/128 (22.7%), direct OAC therapy in 96/128 (75%). All patients with LAA thrombus were under OAC therapy (Vit-K-antagonists in 53.8%, direct OAC in 46.2%).

LAA-thrombus was detected in 26 patients (9 in SR, 17 in AF). Patients with LAA-thrombus (Thrombus group) were older, had higher CHA2DS2-VASc-scores, presented more often with heart failure, and had a more frequent history of stroke/transient ischemic attack (TIA) than those without LAA-thrombus (Control group). Details on patient characteristics are presented in Table 1 and Table 2.

### 4.2. Left Atrial Appendage Thrombus Is Associated with Impaired Mechanical Function

LA-GLS, as measured in transthoracic echocardiography (TTE), differed significantly between thrombus group and the control group, irrespective of the underlying rhythm: 12.4 ± 5.6% vs. 22.1 ± 8.9%, *p* < 0.0001 (Figure 2A). This was also true in the subgroup analysis in the SR and AF cohorts (Figure 2B; LA-LGS for thrombus group vs. control group 14.3 ± 7.4% vs. 24.6 ± 9.0%, *p* < 0.001, and 11.4 ± 4.2% vs. 16.1 ± 5.0%, *p* = 0.045, for SR and AF, respectively).

In line with LA-GLS, LAA-Vel was significantly lower in the thrombus group compared to the control group (19.7 ± 7.5 vs. 41.3 ± 17.5 cm/s, *p* < 0.001; Figure 2C). Moreover, the current study demonstrates a positive correlation between LAA-Vel and LA-GLS (r = 0.39, *p* < 0.001; Figure 2D).

### 4.3. APWD Identifies Patients with LAA-Thrombus with Left Atrial Contractile Dysfunction

A 12-lead-ECG during SR was available in all 128 study patients. Patients in the thrombus group presented significantly longer APWD than those in the control group (202 ms ± 31 ms vs. 153 ms ± 22 ms, *p* < 0.0001; Figure 3A). Application of the previously established threshold of 150 ms for ACM in current study patients [8,9], a total of 81 (63.3%) patients were diagnosed positive for ACM, with 26 (100%) and 55 (53.9%) patients in the thrombus group and control group, respectively (Figure 3B). APWD was inversely correlated to LA-GLS (r = −0.48, *p* < 0.0001; Figure 3C) and LAA-Vel. in TEE (r = −0.39, *p* < 0.0001; Figure 3D).

### 4.4. Diagnostic Value of LA-GLS and APWD for Identification of Patients at Risk for LAA-Thrombus

ROC analysis identified a general threshold of 17.45% for LA-GLS as the best predictor for LAA-thrombus formation in the entire patient cohort (AUC 0.82, 95% CI: 0.73–0.90) with a sensitivity of 84.6% and specificity 63.6% (Figure 4A and Table 3). Rhythm-specific analysis revealed specific thresholds for patients in AF and SR: In AF, a LA-GLS < 13.9% led to an AUC of 0.74 (95% CI: 0.59–0.88) with a sensitivity of 70.6% and specificity of 80.0% (Appendix A). In SR, a LA-GLS < 19.1% led to an AUC of 0.79 (95% CI: 0.66–0.93) with a sensitivity of 77.8% and specificity of 67.1% (Appendix A).

An APWD-threshold of 165 ms was determined by ROC analysis with a high diagnostic value for detection of patients with LAA-thrombus: AUC: 0.90 (95% CI: 0.83–0.97), sensitivity 88.5%, specificity 75.5%, (Figure 3A and Figure 4B and Table 3). Further improvement of the diagnostic performance was observed when LA-GLS and APWD were combined for diagnosis of LAA-thrombus patients, as illustrated in Figure 4C, AUC 0.92 (95% CI: 0.85–0.98).

The diagnostic performance of each threshold is illustrated in Figure 4D and Table 3: 22/26 (84.6%) and 23/26 (88.5%) patients with LAA-thrombus were correctly classified as positive for thrombus when using the LA-GLS threshold 17.45% and the APWD threshold 165 ms, respectively. The combined diagnostic value of both methods is illustrated in Figure 4E and Table 4, with further improvement of sensitivity to detect patients with LAA-thrombus up to 96.2%.

## 5. Discussion

The current study reports two main findings: First, left atrial longitudinal reservoir strain (LA-GLS) and amplified digital p-wave-duration (APWD) identify patients with left atrial cardiomyopathy (ACM) at risk for atrial thrombogenesis. Second, rhythm-specific thresholds for LA-GLS enable identification of ACM patients at risk for LA thrombogenesis during sinus rhythm and atrial fibrillation.

### 5.1. Atrial Cardiomyopathy and Atrial Thrombogenesis

A major role of ACM for LA thrombogenesis and ischemic stroke has been reported in several recent studies: Müller et al. reported that presence of left atrial low-voltage areas in AF patients is associated with history of TIA and stroke [17]. Akoum et al. reported a series of patients with LAA-thrombus who presented LA fibrosis as detected in late Gadolinium-enhanced MRI (LGE-MRI) [18]. Therefore, diagnosis of ACM has been integrated in the recent ESC guidelines on management of patients with AF [19]. We recently reported findings on echocardiographic LA-GLS and electrocardiographic APWD measurement in digital 12-lead-ECG, two non-invasive methods enabling accurate diagnosis of ACM with high (>92%) sensitivity and specificity as compared to left atrial endocardial voltage mapping [9,11]. The current study reveals that use of LA-GLS (in AF or SR) or APWD both allow diagnosis of patients at high risk for LAA-thrombus formation. In addition, the study reports, for the first time, rhythm-specific strain thresholds as well as the APWD threshold for diagnosis of LAA-thrombus in the current cohort of patients with AF.

### 5.2. General and Rhythm-Specific LA-GLS Thresholds for Left-Atrial Appendage Thrombus

LA-strain in SR consists of three components (reservoir strain, conduit strain, and contraction strain), thus achieving a comprehensive assessment of LA contractile function, which has been reported in various studies. LA-strain in AF, on the other hand, was less investigated, as only LA-GLS corresponding to reservoir phase is measurable in this context. In addition, the increased difficulty in identifying the endocardial border in AF has further limited studies on rhythm-specific thresholds. Nevertheless, the difference in LA-strain imaging between SR and AF was observed in the current study (Figure 1A,B). Hereby, awareness should be raised that the inconsistency between SR and AF in contraction patterns and LA-strain imaging reflect the potential underpower of one general threshold for both rhythms in the whole patient cohort, and rhythm-specific thresholds in SR and AF enable further improvement in diagnostic value.

In SR, we found LA-GLS < 19.1% as a potent predictor for LAA-thrombus, which reflects moderate to advanced fibrotic LA remodeling, as a LA-GLS-threshold < 23.5% by previous finding corresponds to early stages of LA fibrotic remodeling/early ACM stages [11]. Notably, the identified LA-GLS threshold in AF of 13.9% for LAA-thrombus corresponds well to the recently identified threshold (13.3%) reported by Liao et al. that identifies AF patients at risk for stroke [14]. This accordance to previous studies strengthens the high reproducibility and the diagnostic value of our current findings.

### 5.3. Combination of Echocardiographic and Electrocardiographic Markers

The study demonstrates that both methods enable risk stratification and identification of patients at risk for LA-thrombogenesis with high accuracy (>84% of patients with LAA-thrombus were identified by both methods and the combination of both methods further improved sensitivity to 96.2%). Moreover, both methods are well correlated to the LAA peak emptying flow velocity in TEE, which has previously been linked to increased risk for LAA-thrombus and ischemic stroke [7].

Nevertheless, albeit the high reproducibility and promising value of APWD in predicting various outcomes was reported in several studies, SR was the prerequisite for this method and the absence of SR in some cases can compromise the diagnostic value of APWD. LA-strain, on the other hand, can be measured both in SR and AF, but demands high image quality and expertise as well as a dedicated strain analysis software. The combination of both markers provides complementary information on the state of structural and functional LA remodeling, yielding an even better diagnostic performance (>96% sensitivity for LAA-thrombus detection) and confirms the diagnosis of ACM by two distinct methods.

## 6. Limitations

The current study included a rather small cohort of 128 patients with AF. One hundred and two patients without LAA thrombus were included as a control group. From a large cohort of patients (1990 cases) undergoing TEE prior to electrical cardioversion, 26 patients with diagnosis of LAA thrombus were included in the current study, in order to assess if the previously reported ACM markers APWD and LA-GLS may be suitable risk markers for LAA thrombogenesis. Establishment of the reported threshold values for both LA-GLS and APWD are based on data collected in these 128 patients. Future large-scaled studies—with inclusion of patient cohorts representing AF patients of all ages and comorbidities—need to refine the diagnostic thresholds for prediction of LAA-thrombus and ischemic stroke with both methods.

## 7. Conclusions

Both LA-GLS by echocardiography and APWD by 12-lead digital ECG enable reliable identification of patients with AF at risk for LA thrombogenesis. Notably, the current study reveals excellent negative predictive values with regard to LAA thrombogenesis for both new ACM markers. Both methods are widely available and may aid clinical decision making in patients with an unclear or borderline risk for LA thrombogenesis using established risk stratification schemes, such as CHA2DS2-VASc-score.

## Figures and Tables

**Figure 1 jcm-11-01332-f001:**
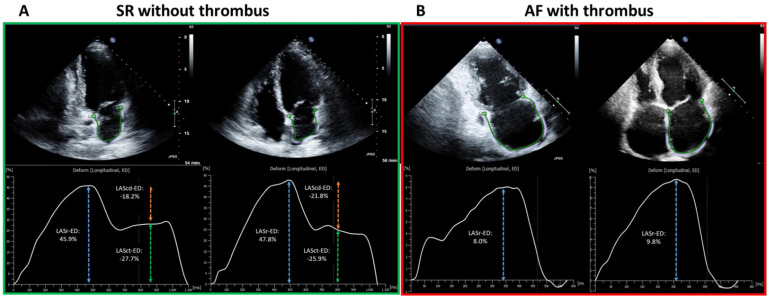
Illustration of LA-Strain measurement in SR (green outlined left panel, (**A**)) and AF (red outlined right panel, (**B**)) from two representative patients. The apical two-chamber and four-chamber view is shown on the left and right, respectively, in each panel. The green curve shows a silhouette of the left atrial endocardium as derived by an automatic border-tracking algorithm. (**A**) demonstrates the TTE image from one patient in SR without LAA-thrombus on TEE, corresponding results of each plane view are depicted as a LA-strain curve with a composite of LASr (reservoir phase, as in blue dashed arrow), LAScd (conduit phase, as in orange dashed arrow) and LASct (contraction phase, as in green dashed arrow). (**B**) shows the TTE image of one patient in AF with LAA-thrombus on TEE. In contrast to SR, LASr remains the only strain parameter for measurement in AF (as in blue dashed arrow). The final result of LA-GLS is calculated as the mean results of LASr from two- and four-chamber view of three consecutive heart cycles (end-diastolic to end-diastolic). LA, left-atrial; SR, sinus rhythm; AF, atrial fibrillation; TTE, transthoracic echocardiography; LAA, left atrial appendage; TEE, transesophageal echocardiography; LASr, left atrial strain in reservoir phase; LAScd, left atrial strain in conduit phase; LASct, left atrial strain in contraction phase; LA-GLS, left atrial global longitudinal strain.

**Figure 2 jcm-11-01332-f002:**
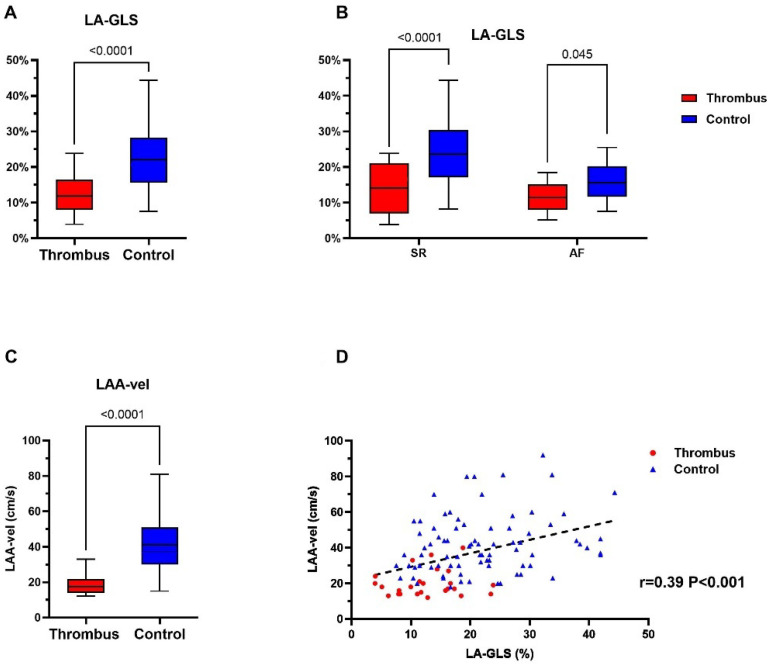
Assessment of echocardiographic parameters for thrombus identification in the thrombus group (red) and control group (blue). (**A**,**B**) demonstrates the difference in LA-GLS between the thrombus and control group in general and rhythm-specific comparison. (**C**) shows the LAA maximal flow velocity in the thrombus and control group. *p* value for each comparison is displayed in each figure (**A**–**C**). (**D**) demonstrates the positive correlation (black dashed line) between LAA-velocity and LA-GLS in individuals of the thrombus- (red) and control-group (blue), respectively. LA-GLS, left atrial global longitudinal strain; LAA, left atrial appendage; LAA-vel, left atrial appendage maximal flow velocity.

**Figure 3 jcm-11-01332-f003:**
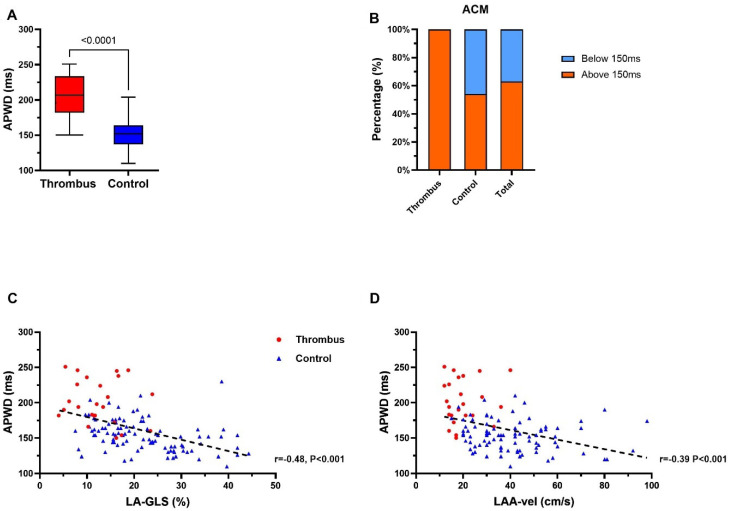
Assessment of electrocardiographic parameter for thrombus and ACM identification in thrombus group and control group. (**A**) demonstrates APWD in thrombus- (red) and control-group (blue). (**B**) illustrates ACM identification using the previously established threshold for detection of ACM of ≥150 ms (APWD below 150 ms (blue) are considered negative for ACM, patients with APWD ≥ 150 ms (red) are considered positive for ACM). (**C**,**D**) display the correlation (black dashed line) between APWD and LA-GLS, and APWD and LAA maximal flow velocity, respectively. ACM, atrial cardiomyopathy; APWD, amplified sinus-p-wave duration; LA-GLS, left atrial global longitudinal strain; LAA, left atrial appendage; LAA-vel, left atrial appendage maximal flow velocity.

**Figure 4 jcm-11-01332-f004:**
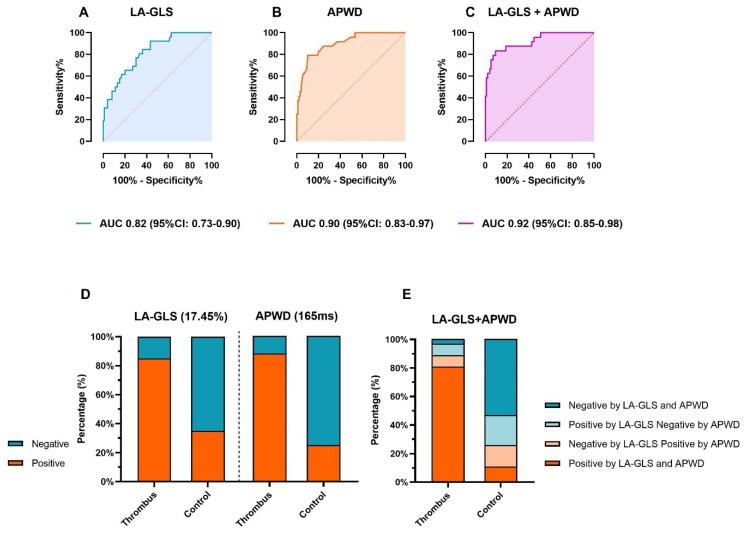
Diagnostic value of LA-GLS and APWD for thrombus identification. (**A**,**B**) show the ROC curves for presence of thrombus for each method. The diagnostic value of the combination of LA-GLS and APWD is illustrated in (**C**). Based on coordinates of ROC curve, a threshold of 17.45% and 165 ms were determined as corresponding thresholds. (**D**) displays the efficacy of each threshold in identifying patients with thrombus. Green and orange bars stand for the percentage of patients that were diagnosed as positive or negative for thrombus based on the corresponding threshold, respectively. (**E**) illustrates the results of thrombus identification by combined thresholds of LA-GLS and APWD, the percentage of patients is illustrated in bar with different colors (orange bars refer to patients who are diagnosed as positive for LAA-thrombus by both LA-GLS and APWD thresholds, green bars refer to patients who are diagnosed as negative by both thresholds, light orange bars refer to patients with one positive diagnosis for LAA-thrombus by APWD and one negative diagnosis by LA-GLS, and light green bars refer to patients with positive diagnosis by LA-GLS and negative diagnosis by APWD. LA-GLS, left atrial global longitudinal strain; APWD, amplified sinus-p-wave duration; ROC curve, receiver operating characteristic curve; LAA, left atrial appendage; AUC, area under the curve; CI, confidence interval.

**Table 1 jcm-11-01332-t001:** Baseline characteristics in each group.

	All Patients (*n* = 128)	Thrombus (*n* = 26)	Control (*n* = 102)	*p* Value
Age (years)	65.8 (10.2)	70.9 (9.2)	64.5 (10.1)	0.004
Sinus rhythm, *n* (%)	82 (64.1)	9 (34.6)	73 (71.6)	<0.001
Persistent AF, *n* (%)	101 (78.9)	26 (100)	75 (73.5)	0.003
Male, *n* (%)	86 (67.2)	18 (69.2)	68 (66.7)	0.804
BMI, kg/m^2^	27.8 (4.1)	18.0 (4.6)	27.7 (4.0)	0.784
Hypertension, *n* (%)	94 (73.4)	22 (84.6)	72 (70.6)	0.148
Diabetes, *n* (%)	16 (12.5)	5 (19.2)	8 (7.8)	0.110
History of Stroke/TIA, *n* (%)	9 (7.0)	5 (19.2)	4 (3.9)	0.017
CHD, *n* (%)	32 (25)	14 (53.8)	18 (17.6)	<0.001
CHA2DS2-VASc-score	2.6 (1.7)	4.2 (1.5)	2.3 (1.5)	<0.001
LAD, mm	44.8 (5.7)	48.8 (6.2)	43.8 (5.2)	<0.001
LAV, mL	90.2 (25.4)	105.6 (17.0)	88.4 (25.6)	0.027
LVEDD, mm	52.8 (6.7)	56.3 (8.6)	51.9 (5.9)	0.003
LVESD, mm	35.7 (9.1)	43.8 (12.0)	33.6 (6.8)	<0.001
LVEF (%)	44.0 (14.9)	29.4 (15.9)	47.9 (11.9)	<0.001
LAA-vel, cm/s	36.9 (17.5)	20.0 (7.5)	42.1 (16.4)	<0.001
Amiodaron, *n* (%)	39 (30.5)	7 (26.9)	32 (31.4)	0.660
Flecanid, *n* (%)	11 (8.6)	0 (0)	11 (10.8)	0.022
Sotalol, *n* (%)	7 (5.5)	0 (0)	7 (6.9)	0.070
Propafenon, *n* (%)	1 (0.8)	0 (0)	1 (1.0)	0.499
Beta-Blocker, *n* (%)	90 (70.3)	22 (84.6)	68 (66.7)	0.074

Persistent AF, persistent atrial fibrillation; BMI, body mass index; TIA, transient ischemic attack; CHD, coronary heart disease; LAD, left atrial diameter; LAV, left atrial volume; LVEDD, left ventricular end-diastolic diameter; LVESD, left ventricular end-systolic diameter; LVEF, left ventricular ejection fraction; LAA-vel, maximal left atrial appendage emptying flow velocity.

**Table 2 jcm-11-01332-t002:** Baseline characteristics in each cohort.

	SR Thrombus (*n* = 9)	SR Control (*n* = 73)	AF Thrombus (*n* = 17)	AF Control (*n* = 29)	*p* Value
Age (years)	66.4 (8)	65.0 (10)	68.2 (9)	68.0 (8)	0.06
Male, *n* (%)	9 (100)	47 (64.4)	9 (47.1)	8 (72.7)	0.09
Persistent AF, *n* (%)	9 (100)	48 (65.8)	17 (100)	27 (93.1)	<0.001
BMI, kg/m^2^	29.3 (4.0)	27.8 (3.9)	27.3 (4.5)	25.3 (2.5)	0.146
Hypertension, *n* (%)	7 (77.8)	52 (71.2)	15 (88.2)	20 (69.0)	0.422
Diabetes, *n* (%)	2 (22.2)	8 (11.0)	3 (17.6)	3 (10.3)	0.041
History of Stroke/TIA, *n* (%)	3 (33.3)	3 (4.1)	2 (11.8)	1 (3.4)	0.05
CHD, *n* (%)	3 (33.3)	14 (19.2)	11 (64.7)	4 (13.8)	0.001
CHA2DS2-VASc-score, *n* (%)	3.44 (1.5)	2.38 (1.5)	4.59 (1.4)	1.9 (1.2)	<0.001
LAD, mm	50.4 (5.8)	43.4 (5.6)	48 (6.4)	44.8 (3.9)	<0.001
LAV, mm	115.3 (9.5)	88.6 (28.1)	99.8 (18.7)	88.0 (21.8)	0.262
LVEDD, mm	61.1 (7.4)	52.7 (5.8)	53.8 (8.3)	49.8 (5.6)	<0.001
LVESD, mm	49.0 (12.4)	33.9 (7.4)	41.1 (11.1)	32.9 (5.2)	<0.001
LVEF(%)	31.2 (17.3)	47.9 (11.8)	28.5 (15.7)	47.7 (12.5)	<0.001
LAA-Vel, cm/s	22.4 (7.5)	41.9 (16.1)	18.6 (7.3)	42.6 (17.5)	<0.001
Amiodaron, *n* (%)	5 (55.6)	25 (34.2)	2 (11.8)	7 (24.1)	0.078
Flecanid, *n* (%)	0 (0)	7 (9.6)	0 (0)	4 (13.8)	0.132
Sotalol, *n* (%)	0 (0)	5 (6.8)	0 (0)	2 (6.9)	0.350
Propafenon, *n* (%)	0 (0)	1 (1.4)	0 (0)	0 (0)	0.770
Beta-Blocker, *n* (%)	6 (66.7)	47 (64.4)	16 (94.1)	21 (72.4)	0.418

Persistent AF, persistent atrial fibrillation; SR, sinus rhythm; AF, atrial fibrillation; BMI, body mass index; TIA, transient ischemic attack; CHD, coronary heart disease; LAD, left atrial diameter; LAV, left atrial volume; LVEDD, left ventricular end-diastolic diameter; LVESD, left ventricular end-systolic diameter; LVEF, left ventricular ejection fraction; LAA-vel, maximal left atrial appendage emptying flow velocity.

**Table 3 jcm-11-01332-t003:** Diagnostic value of thresholds.

Value, %	Sensitivity	Specificity	PPV	NPV
LA-GLS (17.45%)	84.6	63.6	37.9	94.3
APWD (165 ms)	88.5	75.5	47.9	96.2

LA-GLS, left atrial global longitudinal strain; APWD, amplified sinus-p-wave duration; PPV, positive predictive value; NPV, negative predictive value. Diagnostic values of LA-GLS refer to the entire cohort (both rhythms); rhythm-specific ROC analysis is provided in the supplemental data.

**Table 4 jcm-11-01332-t004:** Combination of LA-GLS threshold and APWD threshold for LAA-thrombus diagnosis.

Patients, *n* (%)	Negative by LA-GLS and APWD	Negative by LA-GLS Positive by APWD	Positive by LA-GLS Negative by APWD	Positive by LA-GLS and APWD
Thrombus group, (*n* = 26)	1 (3.8)	2 (7.7)	2 (7.7)	21 (80.8)
Control group, (*n* = 102)	55 (53.9)	15 (14.7)	21 (20.6)	11 (10.8)

Column refers groups of thrombus diagnosis by corresponding threshold; Row refers groups of thrombus detection by TEE; LA-GLS left atrial global longitudinal strain, APWD amplified sinus-p-wave duration.

## Data Availability

The data presented in this study are available on reasonable request from the corresponding author. The data are not publicly available due to privacy restriction.

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
