# Peer review of "Echocardiographic and Electrocardiographic Determinants of Atrial Cardiomyopathy Identify Patients with Atrial Fibrillation at Risk for Left Atrial Thrombogenesis"

_jcm, 2022, doi:10.3390/jcm11051332_

Round 1
Reviewer 1 Report
The manuscript is devoted to the development of clinical decision making method of identifying patients with atrial fibrillation at risk for left atrial thrombogenesis. The aim of the research is to evaluate the corresponding electrocardiographic and echocardiographic predictors by statistical means. The paper consists of problem statement, description of the patient population, methods of statistical analysis, results and discussion. The aim of the research is actual. The methods and datasets are relevant. Experimental design is appropriate, ROC analysis confirmed the proposed method. The research and results are described in detail. The conclusions are based on the results. The references are relevant and mostly up-to-date (5 are out of the last 5 years). On the whole the paper is well-structured and clear.
An issue regarding the large number of self-citations, 7 out of 16 sources refer to the works of the co-authors (sources 3, 4, 5, 7, 8, 9, 13). All of these references are justified but they account for half of all the references.
Author Response
We thank both the editors and the reviewers for their constructive comments on the manuscript, that we would like to adress with this letter. We have accordingly modified our manuscript and we think that with integration of their suggestions, the new version of manuscript has now been improved. We hope it qualifies for publication in the Journal.
Response to Reviewers´Comments:
Reviewer #1:
The manuscript is devoted to the development of clinical decision making method of identifying patients with atrial fibrillation at risk for left atrial thrombogenesis. The aim of the research is to evaluate the corresponding electrocardiographic and echocardiographic predictors by statistical means. The paper consists of problem statement, description of the patient population, methods of statistical analysis, results and discussion. The aim of the research is actual. The methods and datasets are relevant. Experimental design is appropriate, ROC analysis confirmed the proposed method. The research and results are described in detail. The conclusions are based on the results. The references are relevant and mostly up-to- date (5 are out of the last 5 years). On the whole the paper is well-structured and clear.
An issue regarding the large number of self-citations, 7 out of 16 sources refer to the works of the co-authors (sources 3, 4, 5, 7, 8, 9, 13). All of these references are justified but they account for half of all the references..
Answer: We agree that we have integrated in the references a significant number of our previous studies about atrial cardiomyopathy (ACM). The reason for that is that our previous studies are the first detailed reports that have compared non-invasive diagnostic methods for ACM (using digital amplified P-wave analysis and echocardiographic strain analysis) to high-definition invasive electroanatomical voltage maps reflecting presence and degree of ACM in AF patients. The current study is building on these previous results. Therefore, in order to mitigate the extent of self-citations, we have followed your suggestion by adding four additional references that describe the pathophysiology of ACM in the context of AF (ref. 3-6).
Reviewer 2 Report
Major:
-It has been reported that the decrease of the left atrial (LA) strain value is related to the decrease of LA / left atrial appendage (LAA) function and the incidences of LAA thrombus. It is also well known that atrial cardiomyopathy (ACM) is associated with development of AF, LAA thrombo-genesis and stroke.
-In the EHRA / HRS / APHRS / SOLAECE expert consensus (2016) that was cited in Reference 1 in this paper, the definition of ACM is relatively loose, those are "Any complex of structural, architectural, contractile or electrophysiological changes affecting the atria with the potential to produce clinically". In this study, “ACM” was defined by the data including LA enlargement, electrocardiographic conduction delay, and left atrial dysfunction; but it is strongly suggested that the authors have to define the clear criteria of ACM in this text.
-In the AF population, the prevalence of LAA thrombus is usually 3 to 5% when the OACs is properly used; however, in this study, the prevalence of LAA thrombus is as high as 20% (26/128). So, it is essential to state the types of the OACs, the dosage, and the duration of the proper anticoagulant therapy within the context.
-It should be also stated how to enroll the patient. If this study is not a continuous case entering during some study period; it should be stated about the patient selection bias possibilities in the limitation section.
-It would be nice if the baseline characteristics (Tables 1 and 2) describe the type and the duration of AF.
-LA-GLS and APWD should indicate whether they have independent additional value compared to already established useful thrombus prediction scores such as CHADS2, CHA2DS2-VASc scores.
-Although LA-GLS and APWD can be the predictors of LAA thrombus, the sensitivity of LA-GLS and APWD is around 85%, and the prevalence of LAA thrombus is generally 3 to 5%. In the AF cohort, the positive predictive value is expected to be around 10%. Therefore, LA-GLS and APWD could not be very useful predictors of LAA thrombus, but rather may be useful as exclusion factors.
Minor: The spelling of the “CHA2DS2-VASc” score in the text is all inaccurate and should be corrected.
Author Response
We thank both the editors and the reviewers for their constructive comments on the manuscript, that we would like to adress with this letter. We have accordingly modified our manuscript and we think that with integration of their suggestions, the new version of manuscript has now been improved. We hope it qualifies for publication in the Journal.
Response to Reviewers´Comments:
Reviewer #2:
1) It has been reported that the decrease of the left atrial (LA) strain value is related to the decrease of LA / left atrial appendage (LAA) function and the incidences of LAA thrombus. It is also well known that atrial cardiomyopathy (ACM) is associated with development of AF, LAA thrombo-genesis and stroke.
In the EHRA / HRS / APHRS / SOLAECE expert consensus (2016) that was cited in Reference 1 in this paper, the definition of ACM is relatively loose, those are "Any complex of structural, architectural, contractile or electrophysiological changes affecting the atria with the potential to produce clinically". In this study, “ACM” was defined by the data including LA enlargement, electrocardiographic conduction delay, and left atrial dysfunction; but it is strongly suggested that the authors have to define the clear criteria of ACM in this text.
We thank the reviewer for this comment. Indeed, the EHRA consensus paper by Goette et al. Describes different underlying pathohistological findings that are associated with AF. The authors classify these distinct histologies with different subtypes of AF-associated ACM. As atrial histology is not a feasible diagnostic approach in the clinical setting, multiple attempts have been undertaken in the last decade to diagnose left atrial (LA) cardiomyopathy: The first studies revealed presence of late Gadolinium enhancement in MRI, followed by several reports on LA low-voltage substrate (LVS), both of which were shown to predict high arrhythmia recurrence rates after pulmonary vein isolation in Af patients. We recently reported two non- invasive diagnostic methods (APWD and LA-LGS) that allow identification of AF patients with ACM (as identified by presence of LA-LVS>2cm2 in endocardial voltage mapping). Both methods allow diagnosis of ACM with high sensitivity and specificity (>92%). In the current study, we assess if these two non-invasive ACM markers (APWD and LA-GLS) are predictive for LAA-thrombogenesis and report the diagnosis-relevant threshold values. We agree that these aspects were not clearly reported in our manuscript. Following your suggestion, we have now added these background data in the introduction section (page 2, line 46-62, yellow marked sections).
2) In the AF population, the prevalence of LAA thrombus is usually 3 to 5% when the OACs is properly used; however, in this study, the prevalence of LAA thrombus is as high as 20% (26/128). So, it is essential to state the types of the OACs, the dosage, and the duration of the proper anticoagulant therapy within the context.
This is an important remark. As mentioned in the methods section, the current study patients are composed of 102 patients in the control group without LAA-thrombus who underwent TEE prior to their first AF ablation procedure. We specifically, included 26 patients presenting a LAA-thrombus in TEE that was done in a larger
patient cohort (1990 patients) prior to electrical cardioversion for symptomatic AF. Actually, from January 2018 to January 2021, a total of 1990 patients underwent TEE prior to electrical cardioversion at our center. 39 patients had the diagnosis of LAA thrombus in TEE, corresponding to 1,95%. However, LA-GLS was available and measured in 26 of 39 patients. Therefore, the high percentage of events (patients with LAA-thrombus) is due to the fact that the thrombus-patients were specifically included in the current study, in order to assess if APWD and LA-GLS are suitable risk markers for LAA thrombogenesis. We have now mentioned these details in the methods section (page 2 , lines 74-78), as well as in the limitations section (page 10, lines 307-316).
3) It should be also stated how to enroll the patient. If this study is not a continuous case entering during some study period; it should be stated about the patient selection bias possibilities in the limitation section.
As mentioned in response 2, we have integrated these details and limitation in the methods section (page 2 , lines 74-78), as well as in the limitations section (page 10, lines 307-316).
4) It would be nice if the baseline characteristics (Tables 1 and 2) describe the type and the duration of AF.
Following your suggestion, we have now added AF type in patient characteristics table 1 and 2 (page 4 and 5, yellow-marked rows).
However, we are unable to report the AF duration, because this was not systematically documented at our institution. The reason for that is that many patients have uncertainties about the starting date of AF. This information remains uncertain in many cases and is not reportable by some patients who have little symptoms (or who do not know that their fatigue / dyspnea might be related to the arrhythmia and have difficulties even to retrospectively tell the timing of AF beginning or the duration of the longest AF episode).
5) LA-GLS and APWD should indicate whether they have independent additional value compared to the already established useful thrombus prediction scores such as CHADS2, CHA2DS2-VASc scores.
Unfortunately, in the current study we did not include a large representative cohort that reflects all AF patients (comprising all ages and comorbidities) that may be encountered in daily routine. Such a large representative cohort is necessary to assess the diagnostic performance of CHA2DS2-VASc-score and the potential additional value of APWD or LA_GLS (as it was shown by Liao et al (Ref. 14)). Instead of this, because of the rare occurrence of LAA-thrombus (this condition was found in only 1.95% of all patients undergoing TEE prior to planned electrical cardioversion), we preferentially included patients with LAA-thrombus from a large cohort, in order to assess if APWD or LA-LGS (as ACM markers) may be suitable markers for LAA-thrombus. We have now mentioned these limitations in the methods section (page 2, lines 74-78), as well as in the limitations section (page 10, lines 307-316).
CHA2DS2-VASc score please see in the attachment
Following your suggestion, we have calculated the ROC of CHA2DS2-VASc-score ≥2 for our current study cohort (128 patients): We find very similar AUC value of 0.63 for LAA thrombus in our study cohort, as reported in the publications that established this clinical score for stroke risk stratification (AUC of 0.638, Ref. Lip GY, Nieuwlaat R, Pisters R, Lane DA, Crijns HJ. Refining clinical risk stratification for predicting stroke and thromboembolism in atrial fibrillation using a novel risk factor-based approach: the euro heart survey on atrial fibrillation. Chest 2010;137:263-72. and Kirchhof P, Benussi S, Kotecha D et al. 2016 ESC Guidelines for the management of atrial fibrillation developed in collaboration with EACTS. Eur Heart J 2016;37:2893-2962.).
However, because of the non-representative rather small study cohort, our current is a pilot study that identifies new potentially valuable markers for thrombogenesis in AF patients. Our current data are not suitable for a reliable statistical analysis with regard to population-based risk stratofication, as does the CHA2DS2-VASc-score, which was established and evaluated in large population cohorts. We have included the ROC analysis within this response for the reviewer. However, for the above- mentioned statistical reasons we prefer not to include this figure in the manuscript.
.
6) Although LA-GLS and APWD can be the predictors of LAA thrombus, the sensitivity of LA- GLS and APWD is around 85%, and the prevalence of LAA thrombus is generally 3 to 5%. In the AF cohort, the positive predictive value is expected to be around 10%. Therefore, LA-GLS and APWD could not be very useful predictors of LAA thrombus, but rather may be useful as exclusion factors.
We agree with this excellent comment. The exact diagnostic values in the current study for sensitivity, specificity, PPV and NPV for presence of LAA thrombus are reported separately for APWD and LA-GLS in table
- We now mention this important point in the main abstract and in the conclusion of the manuscript (page 10, lines 318-323).
Minor: The spelling of the “CHA2DS2-VASc” score in the text is all inaccurate and should be corrected.
We have now corrected this in the entire manuscript.
